# Experimental and Numerical Investigation on Effects of the Steam Ingestion on the Aerodynamic Stability of an Axial Compressor

**DOI:** 10.3390/e22121416

**Published:** 2020-12-15

**Authors:** Baofeng Tu, Xinyu Zhang, Jun Hu

**Affiliations:** College of Energy and Power Engineering, Nanjing University of Aeronautics and Astronautics, Nanjing 210016, China; zxy970530@163.com (X.Z.); hjape@nuaa.edu.cn (J.H.)

**Keywords:** steam ingestion, axial compressor, total pressure, total temperature, stability margin

## Abstract

In order to investigate the influence of steam ingestion on the aerodynamic stability of a two-stage low-speed axial-flow compressor, multiphase flow numerical simulation and experiment were carried out. The total pressure ratio and stall margin of the compressor was decreased under steam ingestion. When the compressor worked at 40% and 53% of the nominal speed, the stall margin decreased, respectively, by 1.5% and 6.3%. The ingested steam reduced the inlet Mach number and increased the thickness of the boundary layer on the suction surface of the blade. The low-speed region around the trailing edge of the blade was increased, and the flow separation region of the boundary layer on the suction surface of the blade was expanded; thus, the compressor was more likely to enter the stall state. The higher the rotational speed, the more significant the negative influence of steam ingestion on the compressor stall margin. The entropy and temperature of air were increased by steam. The heat transfer between steam and air was continuous in compressor passages. The entropy of the air in the later stage was higher than that in the first stage; consequently, the flow loss in the second stage was more serious. Under the combined action of steam ingestion and counter-rotating bulk swirl distortion, the compressor stability margin loss was more obvious. When the rotor speed was 40% and 53% of the nominal speed, the stall margin decreased by 6.3% and 12.64%, respectively.

## 1. Introduction

The carrier-based aircraft is a kind of aircraft that can eject, take off and land on an aircraft carrier. During the steam ejection takeoff process, the high-temperature steam leaked from the steam ejection system of the carrier will be ingested by the aircraft engine again [1,2,3,4,5,6]. When the hot gas, which is composed of steam and air, is ingested by the engine, a stall may occur in the compressor ahead of time [7].

During the period from 2004 to 2014 [7,8,9,10,11,12,13], the naval Graduate School in California conducted many experiments and numerical simulations on a transonic rotor compressor with steam ingested. The experimental results show that as the power of the turbine driving the compressor was constant, the rotor speed was forced to increase under the intake of steam into the compressor, and the angle of the attack of the blade was changed; thus, the compressor stability margin was reduced significantly, and the compressor entered instability state. With the increase of the rotational speed, the stall margin decreased. The forward-swept rotor compressor was also tested. The results displayed that a certain forward-swept blade profile could reduce the adverse effect of steam ingestion on the compressor stall margin. In 2009, Zarro [10] carried out the steady numerical simulation research on a single rotor channel of a transonic compressor with the ANSYS CFX software. It was found that the calculation method employing the k-ε turbulence model received more reasonable results. In the calculation domain, a gas mixture of ideal air and ideal water gas was used as the inlet working fluid. The numerical simulation results show that the compressor stability boundary moved to the right obviously under steam ingestion, and the compressor stall margin was reduced. In addition, the shock wave in the transonic rotor was moved upstream due to steam ingestion. The further the shock wave moved upstream, the more likely the compressor was to enter the stall state. Therefore, the position change of the shock wave caused by steam ingestion may be the main reason that affected the aerodynamic stability of a transonic compressor. In 2014, Gannon [13] conducted a comprehensive experimental study on the stalling characteristics of a single-spool transonic compressor. It was found that steam ingestion decreased the stall margin of the transonic compressor. When the mass and volume fraction of steam were low in the experiment, the physical properties of the mixture of steam and air changed little during the steam ingestion. Steam ingestion led to a faint rise in the intake flow temperature and the velocity of sound. Though steam had a limited influence on the mixture gas properties and velocity of sound, it did deteriorate the compressor stability.

Tu [14,15] studied steam ingestion on the transonic rotor 37 and a low-speed axial-flow compressor by numerical simulation. The results show that steam reduced the dimensionless stable operating range of a compressor. The more the intake amount of steam was, or the higher the steam temperature was, the more the compressor stable operating range decreases. Moreover, in the condition of the same intake steam mass fraction, the stable operating range at a high rotor speed decreased more than that at a low rotor speed. The effect of temperature rise rate on the stability boundary was not notable. Further research shows that under the same intake amount of steam, with the increase of the high-temperature steam ingestion area, the total temperature of the region decreased, and the stable operating range was extended. With the increase of the ingestion area, the change of the stable operating range became smooth. Based on the parallel compressor model, Xu and Hu [16] studied the influence of gas properties on compressor characteristics. The results show that compared with the uniform inlet case, in the steam ingestion case, the mass flow at the stability boundary point was larger, while compressor stability margin and the pressure ratio characteristic parameters were much less under the same total temperature.

In terms of numerical calculation, the existing numerical calculation only adds ideal air and ideal water gas into the calculation domain without considering the coupling calculation of energy, momentum and mass exchange among air, gas vapor and water. In practice, the steam leaked from the aircraft carrier deck appeared to be white, indicating that the steam contained liquid water. In the present work, the multiphase flow model was first applied to study the effect of steam ingestion on the aerodynamic stability of a two-stage axial-flow compressor. The coupling calculation of energy, momentum and mass exchange among the three elements of air, gas vapor and liquid water was conducted in the numerical simulation. The numerical calculation method was verified by the experimental study of steam suction in a compressor. Therefore, the proposed numerical approach is a more accurate reflection of the experimental condition.

In addition, when a carrier aircraft adopts an S-shaped inlet, it will be affected by steam suction and swirl distortion at the same time when taking off. Swirl distortion also influences the compressor aerodynamic stability [17,18,19,20,21], and it may further deteriorate the compressor stability when combined with steam suction.

In this paper, a two-stage low-speed axial-flow compressor was taken as the research object. A test-bed was established to simulate the environment of steam and swirl distortion for the compressor. Mere steam suction and steam suction with swirl distortion were carried out to study the effects of steam ingestion and swirl distortion on the aerodynamic performance and stability of a compressor.

## 2. Experimental Equipment

The experimental research was carried out on a two-stage low-speed axial-flow compressor test bench, which was composed of a steam generator, a bell mouth, an intake pipe, a compressor, a volute, an exhaust pipe, an electric valve, a muffler, a 200 kW DC motor and a speed controller (Figure 1).

The designed rotational speed of the compressor was 1500 r/min. The rotor and stator used NACA-65-010 blade type, and the inlet guide was not installed. The overall structure layout was an axial inlet and radial exhaust. The detailed data are presented in Table 1.

By injecting steam into the inlet pipe of the compressor and installing a swirl distortion generator at the compressor inlet, the effects of steam ingestion and swirl distortion on compressor performance and aerodynamic stability were studied.

The maximum load-power of the electric heating steam generator was 144 kW, the maximum evaporation capacity was 200 kg/h, and the maximum steam temperature was 443.15 K. The steam produced by the steam generator contained small liquid droplets, and the maximum temperature referred to the temperature of the gaseous water vapor in the steam. The generator had a long steam inlet pipe and an expanded outlet so that steam and air were fully mixed in the inlet pipe, and the inlet air was uniform in the circumferential direction of the compressor inlet.

The swirl distortion was produced by the blade-type swirl distortion generator (BSDG) designed by Tu [22]. The BSDG was located at about 2.9 times the outer diameter upstream of the first stage rotor, as shown in Figure 2. The swirl distortion generator could produce co-rotating bulk swirl and counter-rotating bulk swirl (the inlet swirl was in the opposite direction of rotor revolution).

In the experiment, the swirl distortion generator was used to produce a counter-rotating bulk swirl, which could increase the total pressure ratio of the compressor and reduce the stable working range of the compressor so as to simulate the swirl distortion produced by an S-shaped inlet.

The mass flow, total pressure ratio, and efficiency of the compressor were measured in the experiment. The overall arrangement of the measuring probes is represented in Figure 3.

At the 0–0 cross-section, four wall static pressure measurement holes were uniformly arranged in the circumstance to obtain the average static pressure at the compressor inlet and determine the mass flow. At the 1–1 cross-section, six six-point total pressure rake probes were uniformly arranged in the circumstance to measure the total pressure at the inlet of the compressor’s first rotor. At the 2–2 cross-section, sixteen static pressure holes were distributed along the circumstance on the inner and outer casing to measure the average static pressure at the outlet of the second stator. At the 3–3 cross-section, eight six-point total pressure rake probes were arranged in the circumstance to measure the total pressure at the compressor outlet. The ambient temperature and pressure were measured using the experimental mercury thermometer and mercury pressure gauge.

The steady-state acquisition system consisted of a JC torque measuring instrument, a rotational speed measuring instrument, a PSI DTCnet Acq pressure measuring system, and a computer. The JC torque measuring instrument was used to measure torque, rotational speed and motor power, and its torque and rotational measurement accuracy were up to 0.2% and 0.1%, respectively. The DTCnet Acq pressure measuring system and the computer that recorded data were used to measure the PSI (pounds per square inch, 1 PSI = 6.895 kPa) of the steady pressure. The system has a function of digital temperature compensation. The sampling rate was 400 Hz per passage. The two kinds of steady-state pressure measuring sensors were adopted in all experiments mentioned in the paper. The range was 1 PSI (6.895 kPa) and 2.5 PSI (17.237 kPa), respectively. The measurement accuracy was ±3.45 Pa and ±8.62 Pa, respectively.

## 3. Experimental Method

The experiment was conducted through the following three steps. First, feed the compressor with uniform inlet, then the reference characteristic map of the total pressure ratio was measured, and the position of compressor stability boundary point was determined. Second, the compressor was set to operate at the maximum opening position of the valve at a specific rotational speed, and then the steam with set parameters was continuously injected into the compressor. Steadily reduce the valve opening to allow the compressor to work at the near stall point, and determine the characteristic parameters. Change the rotational speed and repeat the current step. Third, attach the swirl generator to the compressor, and repeat the first two steps. The stability boundary point of the compressor was determined by observing the fluctuation of the characteristic data. The stall type was an abrupt stall. The sudden drop of the total pressure ratio at the small mass flow working state on the characteristic line indicated the occurrence of the stall in the compressor.

Due to the high electric power consumption of the steam generator, the motor power allocated to the compressor was not enough to run the compressor at the design speed. In the experiment, the compressor was tested under the rotational speed of 40% and 53% of nominal speed.

In order to highlight the change of inlet total temperature caused by steam ingestion, the existence of a few liquid tiny-diameter water droplets in the steam was ignored, and steam was regarded as gaseous water vapor, which was a kind of ideal gas. The air temperature was 288.15 K, and the specific heat at a constant pressure of the air was 1004.4 J/(kg·K); the steam temperature was 416.75 K, and the specific heat at a constant pressure of steam was 1900.64 J/(kg·K). The mixing process was assumed to be a constant pressure process. The mass flow of steam was 150 kg/h, and the mass flow was fixed in the experiment. When the compressor was working at the stability boundary point at 40% of the nominal speed, the physical mass flow was 7.2 kg/s, and the mass fraction of steam was 0.58%.

The specific heat at a constant pressure of mixed gas was calculated by the following formula:(1)$$C=w_{a}C_{a}+w_{s}C_{s}=1009.6\;J/(\mathrm{kg}\cdot K)$$
where *C* is the specific heat at constant pressure, *w* is the mass fraction, *a* is air, and *s* is steam.

According to the conservation of energy,
(2)$$C_{a}T_{a}^{*}m_{a}+C_{s}T_{s}^{*}m_{s}=CT^{*}m$$
where $T^{*}$ is the total inlet temperature, and *m* is the mass flow.

The total temperature of the gas mixture
(3)$$T^{*}=\frac{C_{a}T_{a}^{*}m_{a}+C_{s}T_{s}^{*}m_{s}}{C_{m}}=289.6\;K$$


At 53% of the nominal speed, the physical mass flow at the stability boundary point was 9.7 kg/s, and the total inlet temperature of mixed gas was 289.2 K.

Temperature distortion intensity is defined as follows:(4)$$D_{T^{*}}=\frac{T_{\mathrm{av},\mathrm{HI}}^{*}-T_{\mathrm{av}}^{*}}{T_{\mathrm{av}}^{*}}$$
where $T_{\mathrm{av}}^{*}$ is the surface average temperature, and $T_{\mathrm{av},\mathrm{HI}}^{*}$ is the surface average temperature of the high-temperature area. The total temperature distortion intensity $D_{T^{*}}$ was 0.5% at 40% of nominal speed and 0.36% at 53% of nominal speed. The increase of rotational speed led to the increase of airflow, thus decreasing the steam mass fraction and the total temperature distortion intensity.

The corrected mass flow was calculated according to the following formula:(5)$$m_{1,\mathrm{cor}}=m_{1}\sqrt{\frac{T_{1}^{*}}{288.15}}\frac{101325}{P_{1}^{*}}$$
where $m_{1,\mathrm{cor}}$ is the corrected mass flow, $m_{1}$ is the physical mass flow, $T_{1}^{*}$ is the total inlet temperature, and $P_{1}^{*}$ is the total inlet pressure. The flow in the subsequent numerical simulation was also converted to the corrected mass flow according to this method.

Stall margin of the compressor is defined as follows:(6)$$SM={\left[\frac{{\left(\frac{\pi^{*}}{m_{1,\mathrm{cor}}}\right)}_{b}-{\left(\frac{\pi^{*}}{m_{1,\mathrm{cor}}}\right)}_{e}}{{\left(\frac{\pi^{*}}{m_{1,\mathrm{cor}}}\right)}_{e}}\right]}_{n_{\mathrm{cor}}=\mathrm{const}}\times 100\%$$
where SM is the stall margin, $\pi^{*}$ is the total pressure ratio, e is the highest efficiency working point, b is the stability boundary point, and $n_{cor}$ is the corrected speed. The highest efficiency point (the maximum mass flow point as well) of this type of compressor was achieved under the full opening of the valve.

## 4. Experimental Results

Four types of experiments were carried out, including uniform inlet, steam ingestion, swirl distortion, and the combination of steam ingestion and swirl distortion (Figure 4). Legends of “air”, “st”, “sw” and “st+sw” in the figure indicate the uniform inlet test, the steam ingestion test, the swirl distortion test, and the combination of steam ingestion and swirl distortion test at each rotational speed. The baseline instability boundary is the collection of the last stable operating points before the compressor enters the stall state. On the right side of the baseline stability boundary are the compressor stable boundary (swirl-induced stability boundary line) for testing swirl distortion, the compressor stable boundary for testing steam ingestion (steam-induced stability boundary line), and the compressor stable boundary (swirl- and steam-combined-induced stability boundary line) for testing the combination of steam ingestion and swirl distortion. To show the difference between numerical simulation and experiment results more directly, the parameter changes are described in percent in detail in the following subsections.

### 4.1. Steam Ingestion

Table 2 displays the test results of the influence of steam ingestion on the compressor stall margin. Table 3 and Table 4 show the decrease values of the peak total pressure ratio and the peak efficiency on the characteristic line. Under the condition of the uniform inlet, the compressor stall margin decreased with the increase of the speed. The stall margin decreased by 1.5% and 6.3% when the compressor speed was 40% and 53% of the nominal speed, respectively. Therefore, the higher the rotational speed was, the more significantly the stall margin was reduced. When the compressor speed was 40% and 53% of the nominal speed, the peak total pressure ratio decreased by 0.0064% and 0.0196%, and the peak efficiency decreased by 1.54% and 2.03%, respectively. Since the design pressure ratio of the compressor was relatively small, the variation range of the maximum total pressure ratio was also trivial. However, the changing trend was consistent; that is, the higher the rotor speed was, the more the maximum total pressure ratio and the peak efficiency decreased. At the same time, with the decrease of mass flow, the characteristic difference of small mass flow also increased. The main reason is that the decrease of air mass flow gave rise to the increase of the proportion of steam mass flow; thus, the negative effect of steam ingestion on the compressor aerodynamic stability was more serious.

### 4.2. Swirl Distortion

The variations of the compressor stall margin, maximum total pressure ratio and peak efficiency are given in Table 5, Table 6 and Table 7, respectively. The reference stall margin and the maximum total pressure ratio were obtained by the uniform inlet condition. The swirl distortion generator produced a counter-rotating bulk swirl. Under the counter-rotating bulk swirl, the angle of attack of the rotor increased, and the blade was more loaded. As a result, the compressor pressure ratio increased, and the stall margin decreased. The higher the rotational speed was, the more significant the change of total pressure ratio and the stability margin were. When the compressor rotational speed was 40% and 53% of the nominal speed, the stall margin decreased by 3.58% and 6.4%, the maximum total pressure ratio increased by 0.0032% and 0.0160%, and the peak efficiency decreased by 1.89% and 2.14%, respectively. With the increase of the rotor inlet angle of attack, there was an increasing tendency of boundary layer separation on the suction surface. The aerodynamic load of the blade increased, and the compressor characteristic line moved towards the direction of a higher pressure ratio and mass flow, and the stall margin decreased.

### 4.3. Steam Ingestion and Swirl Distortion

The variations of the stall margin, the maximum total pressure ratio and the peak efficiency are given in Table 8, Table 9 and Table 10. When the compressor speed was 40% and 53% of the nominal speed, the stall margin decreased by 6.3% and 12.64%, the maximum total pressure ratio decreased by 0.0065% and increased by 0.0015%, and the peak efficiency decreased by 3.32% and 4.06%, respectively. Under the combination of swirl distortion and steam ingestion, the compressor stall margin decreased more than the case where either one of them worked alone, and the stall margin loss had a rising trend with the increase of the compressor rotational speed. The swirl distortion resulted in the increase of the inlet angle of attack, the increase of total pressure ratio and the decrease of the stall margin; while the steam ingestion reduced the compressor corrected speed, inlet Mach number and mass flow, thus decreasing the total pressure ratio and the stall margin. When the compressor was working at the large mass flow range, the total pressure ratio was larger than the reference total pressure ratio, and the opposite is true when the mass flow was small. The results show that the proportion of the steam mass flow at the working point with the small mass flow was more than that at the working point with the larger mass flow. The influence of steam ingestion on compressor total pressure ratio was greater than that of swirl distortion.

## 5. Numerical Simulation Method

Since steam entered the compressor inlet, energy, momentum, and mass exchange between steam and air or water droplet began. In the compressor, steam and hot liquid water released heat, while air absorbed the heat, and its temperature and pressure were increased. In this paper, the multiphase flow calculation method based on the Euler–Lagrange multiphase flow model was used in the full passages model of a two-stage axial-flow compressor. The ideal air, ideal gaseous water and liquid water were defined in the calculation domain. Through the coupling calculation of continuous phase and discrete phase, combined with steady numerical simulation methods, the influence of steam on the compressor aerodynamic stability was studied.

The governing equations of the continuous phase, which contains air and gaseous water, are solved by the Eulerian method. The mass, momentum, and energy conservation equations are presented as follows:

The first is the mass conservation equation.
(7)$$\frac{\partial \rho }{\partial t}+\nabla \cdot \left(\rho \vec{u}\right)=S_{m}$$
ρ and $\vec{u}$ are the density and velocity of mixed gas, respectively. The source term, $S_{m}$, represents the mass transfer from gas to droplet, or discrete phase to the continuous phase.

The second is the momentum conservation.
(8)$$\frac{\partial }{\partial t}\left(\rho \vec{u}\right)+\nabla \cdot \left(\rho \vec{u}\vec{u}\right)=-\nabla p+\nabla \cdot \left(\overset{\bar{¯}}{\tau }\right)+\vec{F}$$
p is the static pressure, $\vec{F}$ is the external volume force, and $\overset{\bar{¯}}{\tau }$ is the viscous shear stress tensor.

The third is the energy conservation equation.
(9)$$\frac{\partial }{\partial t}\left(\rho h_{t}\right)+\nabla \cdot \left(\vec{u}\rho h_{t}+p\right)=\nabla \cdot \left(\lambda \nabla T+\left(\bar{\tau }\cdot \vec{u}\right)\right)+\vec{u}\cdot \vec{F}+S_{h}$$
$h_{t}$ is the total enthalpy, λ is the thermal conductivity, and $S_{h}$ is the heat source term.

The discrete phase governing equations of liquid droplets in steam were solved by the Lagrangian method. The governing equations include the discrete phase particle motion equation, the heat transfer equation, and the mass transfer equation. The specific equations are shown below.

The first is the motion equation of the liquid droplet.
(10)$$m_{p}\frac{d\vec{u_{p}}}{dt}=\vec{F_{D}}+\vec{F_{B}}+\vec{F_{R}}+\vec{F_{VM}}+\vec{F_{P}}$$
$m_{p}$ is the mass of liquid droplets, $\vec{u_{p}}$ is the velocity of the liquid droplet,$\;\vec{F_{D}}$ is the aerodynamic drag on the liquid droplet, $\vec{F_{B}}$ is the buoyancy item, $\vec{F_{R}}$ is the centrifugal force imposed on the liquid droplet, and $\vec{F_{VM}}$ and $\vec{F_{P}}$ are virtual mass force and pressure gradient force, respectively.

In Lagrangian coordinates, the droplet motion equation can be simplified as follows:(11)$$m_{p}\frac{d\vec{u_{p}}}{dt}=\vec{F_{D}}+\vec{F_{R}}$$
(12)$$\vec{F_{D}}=C_{D}\left(\vec{u}-\vec{u_{p}}\right)$$
(13)$$\vec{F_{R}}=m_{p}\left[-\vec{\omega }\times \left(\vec{\omega }\times \vec{r}\right)-2\vec{\omega }\times \vec{u_{p}}\right]$$
where $\vec{\omega }$ is the rotational speed, and $C_{D}$ is the drag coefficient.

Second, the heat transfer equation.
(14)$$mC\frac{dT}{dt}=-\left(\pi d_{p}\lambda Nu\left(T-T_{P}\right)+\pi d_{a}\lambda Nu\left(T-T_{a}\right)+\frac{dm}{dt}h_{fg}\right)$$
C is the specific heat of the continuous phase, and λ is the thermal conductivity of the surrounding continuous phase gas. T, $T_{p}$ and $T_{a}$ are the temperature of gas steam, droplet and air, respectively.$\;d_{p}$ and $d_{a}$ are the diameter of the water droplet particle and air particle, respectively. Nu represents the Nusselt number and $h_{fg}$ is the latent heat of evaporation.

Third, the mass transfer equation.

Using the Antoine equation to determine the saturation pressure.
(15)$${log}_{10}p_{sat}=A-\frac{B}{T+C-273.15}$$
The coefficients of A, B and C were empirical constants [23].

The mass transfer is related to the gas phase temperature in the phase transformation model. In this paper, the temperature of the gaseous water is higher than the temperature of the phase transition point, and the phase change rate is determined by the convective heat transfer.
(16)$$\frac{dm}{dt}=\frac{\pi d_{p}\lambda Nu\left(T-T_{p}\right)+\pi d_{a}\lambda Nu\left(T-T_{a}\right)}{h_{fg}}$$

The two-way coupling (volume fraction coupling and temperature coupling) between discrete and continuous phase are realized by adding term source in the respective phase governing equations.

In the present work, a two-stage low-speed axial compressor model was numerically simulated, and the specific design parameters of the compressor are represented in Table 1. The whole loop parallel computing was used in the numerical calculation. The Autogrid5 module of the NUMECA software package was used to generate the single-channel grid automatically (Figure 5). The computational grid used in the full passage calculation was generated by copying the single-channel grid in the circumferential direction in the CFX software. The blade channel adopted the HOH structured grids. The single-channel grid of the two-stage axial compressor was about 700 thousand, and the total grid amount of the full passages was about 14 million (Figure 6). Grid numbers of 12 million, 14 million and 16 million were selected to verify the grid independence. The numerical calculation data at 53% of nominal speed was compared with the test data (Figure 7). The legends of 12 million, 14 million and 16 million in the figure represent the numerical calculation results of different grid numbers, respectively, and experience means the calculation results of the mere air ingestion test. There was a big gap between the 12 million grids and the test results, while the difference between the calculation results of 14 million grids and 16 million grids was reasonable. The total pressure ratio of numerical calculation was about 1% lower than the test value, and the efficiency was about 1% lower. The difference between the numerical calculation and the experiment was acceptable, and the accuracy of the numerical simulation was guaranteed. Taking into account the computing power and time of the computer, the grid number of 14 million was finally selected.

In this paper, the CFX software was used for numerical simulation. The calculation was carried out at 40% and 53% of the nominal speed, respectively, to obtain a stable boundary line. In the computational domain, the mixture of ideal air, gaseous water and liquid droplet was selected. Because the volume fraction was used instead of mass fraction in the multiphase flow calculation, gaseous water and air were regarded as an ideal gas, and the volume of an ideal gas in 1 mol was the same in the standard state. When the rotational speed in the calculation domain was set at 40% and 53% of the nominal speed, the mass fraction of steam in all imported materials was set as 0.5% and 0.38%, respectively, thus the steam intake volume was basically close to that in the experiment. Considering that the steam generated by the steam generator contained water droplets, the mass fraction of liquid water was set as 1/10 of the gaseous steam, and the rest was ideal air. The diameter of the water droplet was set as 5 μm.

For the favorable robustness, low computation cost and high universality, the standard k-ε turbulence model is viewed as a normal industrial turbulence model in the industry community since its inception, and it is widely applied in practical flow and heat transfer simulation. Standard k-ε turbulence model based on previous research paper [13] on the compressor wet compression and steam ingestion was applied, and the method of scalable wall functions was used near the wall. The Y+ value range of rotor blades and stator blades was 0.45 to 3.01 and 0.71 to 2.38, respectively.

The computational domain is represented in Figure 8, and boundary conditions are represented in Table 11. In the calculation process, the airflow direction was axial, and the hub, shroud and blades were both adiabatic, smooth and non-slip. At the inlet, the pressure was 101.325 kPa, the air temperature was 288.15 K, the steam temperature was 416.15 K, the water temperature was 363.15 K, and the critical stagnation temperature of the steam phase transition was 373.15 K. Static pressure was given as an outlet boundary condition.

In this paper, the steady-state simulation was used to study the effect of steam ingestion and swirl distortion on the stability of the two-stage compressor at different speeds. In the steady-state simulation, the static pressure was given as the outlet boundary condition, and the effect of the throttle valve was simulated by increasing the static pressure value to obtain a complete characteristic line. Computations were performed with the frozen rotor approach. The calculated convergence criterion was set as 1 × 10^−5^ RMS. It took 64 CPUs 6 h to calculate a steady-state example. In order to verify the accuracy of the steady simulation, the characteristics of the two-stage compressor were calculated with mere air intake. The total temperature and pressure at the compressor inlet were 288.15 K and 101,325 Pa, respectively. The characteristic curve is shown in Figure 9. The characteristic curves of the steady simulation were consistent with experiment results, which shows the reliability of the calculation results.

## 6. Numerical Simulation Results

Figure 10 shows the total pressure ratio vs. mass flow and efficiency vs. mass flow characteristics of the two-stage compressor employing the full-passage calculation model. The legend “air-cal”, “st-cal”, “air-exp” and “st-exp” in the figure represent the numerical calculation reference characteristic line, steam ingestion numerical calculation characteristic line, experimental reference characteristic line, and steam ingestion experimental characteristic line at each rotational speed. The baseline stability boundary of the compressor for mere air ingestion experiment from 40% to 53% of nominal speed is marked as “baseline stability boundary”. On the right side of the baseline stability boundary are the compressor baseline stability boundary obtained from CFD, steam induced stability boundary from experiment and steam induced stability boundary from CFD.

Table 12, Table 13 and Table 14 show the results of the numerical calculation and the experiment stability margin, maximum total pressure ratio and peak efficiency reduction of the two-stage compressor under steam ingestion. The stall margin and the maximum total pressure ratio of the numerical calculation went down even more than the experimental results. In the numerical calculation, when the compressor rotational speed was 40% and 53% of the nominal speed, the stall margin was reduced by 3.21% and 8.12%, the maximum total pressure ratio was reduced by 0.0080% and 0.0216%, and the peak efficiency was reduced by 1.97% and 2.31%, respectively. Taking the rotational speed of 40% of nominal speed as an example, the numerically calculated stall margin, maximum total pressure ratio and peak efficiency were 1.71%, 0.0016% and 0.43% higher than the experimental values, respectively. The difference can be attributed to the numerical calculation model setting that the wall was defined as an adiabatic wall, which ignored the effect of steam condensation on itself. Although the results were not consistent, the trend of stability margin and total pressure ratio was similar, that is, the higher the compressor rotational speed was, the more significantly the stability margin and maximum total pressure ratio decreased.

Figure 11 shows the static temperature distribution of the full-passage calculation model of the two-stage compressor under two different conditions of near stall point at 98% blade height, including the steady calculation results with uniform inlet and steam ingestion. The static air temperature of the flow increased continuously from the inlet of the compressor due to steam ingestion. After the flow pass through the second stage rotor passage, the air static temperature increased because of energy exchange between steam and air. The phenomenon of heat release from steam to air existed in the whole flow process.

Figure 12 and Figure 13 show the distribution of limiting streamline on the suction surface of the blade under uniform inlet and steam ingestion conditions at the near stall point. For different circumferential positions, three channels (passage 1, passage 2 and passage 3) were selected. The flow conditions of the three channels at different circumferential positions were the same when the inlet air was uniform. The distribution of limiting streamline on the suction surface of Passage1 was given as representative (Figure 9). Under the condition of steam ingestion, due to the unsteady flow field, there were certain differences in the flow conditions at different circumferential positions. Therefore, the distribution of limiting streamline on the suction surface of the three channels is given (Figure 10). The radial movement of the limiting streamline on the suction surface of the blade revealed the size of the flow separation area, and it was also a sign of the formation of the separation vortex. The results show that the flow separation occurred in the corner region at the lower end of the trailing edge of the first stage rotor blade suction surface under the uniform inlet condition, and the separation zone occupied nearly 20% of the blade height. The flow separation region at the suction surface of the second stage rotor blade was smaller (occupied nearly 10% of blade height). Under the condition of steam ingestion, the separation region at the suction surface of the first and second stage rotor blades approximately owned the whole blade height. The separation was more serious than that of the uniform inlet case. The separation at the suction surface of the second stage rotor blade was more serious than that of the first stage rotor blade, indicating that the influence of steam ingestion on the second stage rotor was more pronounced. This may be due to the continuous heat transfer between steam and air in the compressor passage. The heat transfer led to a temperature rise of air. The corrected speed of the compressor under the same physical speed increased, the mass flow rate decreased, and the separation process was promoted to take place. Therefore, entropy and flow loss increased significantly in the passage of the rear stage.

The distribution of air Mach number in the full passages at 10% blade height under the condition of uniform inlet and steam ingestion was analyzed (Figure 14). The Mach number in the compressor passages during steam ingestion was lower than that in the uniform inlet case, indicating that steam ingestion reduced the air mass flow and the stall margin. The thickness of the boundary layer and the area of the low-speed region on the suction surface of most blades at all stages were both increased, and the wake loss was also increased. Under the action of the reverse pressure gradient, the thickness of the boundary layer on the suction surface increased, and ultimately flow separation and compressor stall occurred.

The distribution of entropy reveals the distribution of flow loss in blade rows. In Figure 15, the phenomenon of entropy production at the trailing edge of the blade reveals an increase in wake loss in certain areas. If the flow loss in the blade passage or on the blade wall is too severe, flow separation may occur, which deteriorates the compressor stability. Figure 15 shows the distribution of air static entropy in the full passages at 98% blade height under the condition of uniform inlet and steam ingestion, and the flow loss on the blade wall and in the passages is reflected.

At the near stall point, the entropy value in the passages affected by steam ingestion was significantly higher than that in other passages, and the entropy was high in the whole passage; thus, the flow loss was severe. The entropy increase was noteworthy in the rotor and stator passages, especially in the wake region of the second stage stator trailing edge and tip leakage flow region, indicating that the flow loss in the passages and the tip leakage flow region was increased due to steam ingestion. The flow stability in the compressor was also reduced because of the existence of these high entropy regions.

Figure 16 shows the distribution of water volume fraction on the blade wall under the condition of steam ingestion. From left to right, the first stage rotor blade (R1), the first stage stator blade (S1), the second stage rotor blade (R2), and the second stage stator blade (S2) are corresponding. The upper part is the suction surface of each blade, and the lower part is the pressure surface of each blade. The volume fraction of water represents the size of the water film on the wall. The majority of the front edge of the suction surface of the R1 blade was completely covered by water film (shown in green and yellow), and the phenomenon indicated that incoming water droplets first hit the first stage rotor. A small part of the S1 blade trailing edge was covered by water film. The water film adhered to the wall (shown in pale blue), thus reducing the kinetic energy of incoming air and vapor and ultimately reducing Mach number. A small scope of the R1 pressure surface was attached by water film. There was a large range of water film on the front edge of the R2 blade. Due to the influence of gravity, the water droplets in the passage of the rear stage moved towards the blade root and the bottom casing. The rear stage was impacted by fewer water droplets; hence almost no water film adhered to the pressure and suction surfaces of the S2 blade.

From the above analysis, it can be inferred that steam ingestion changed the flow condition in the compressor passages. The first stage rotor of the compressor was affected by steam first, and the second stage passage was more seriously affected by steam than the first stage. The results show that the steam ingestion reduced the incoming Mach number, decreased the kinetic energy, and thickened the boundary layer of most of the rotor suction surface, thus resulting in the expansion of the flow separation area and increase of the wake flow loss. The boundary layer on the suction surface of the rotor was easier to separate under the condition of small mass flow compared with the case of the uniform inlet, which made the compressor instability start early.

## 7. Conclusions

In this paper, the influence of steam ingestion and swirl distortion on the aerodynamic stability of a two-stage low-speed axial-flow compressor was investigated through experiment and multiphase flow numerical simulation, and the influence of swirl distortion on the compressor stability was studied by experimental method.

Steam ingestion reduced the compressor’s stable operating range, and the higher the rotational speed, the more the stall margin was reduced. When the rotor speed was 40%, and 53% of nominal speed, the compressor stall margin decreased by 1.5% and 6.3%, respectively. The stall margin was decreased under the counter-rotating swirl distortion generated by the swirl distortion generator, and the higher the rotational speed, the greater the negative effect of the swirl distortion on the compressor stall margin. When the rotor speed was 40% and 53% of the nominal speed, the stall margin decreased by 3.58% and 6.4%, respectively. The combined effect of steam ingestion and counter-rotating swirl distortion resulted in a more significant decrease in the compressor stall margin. When the compressor speed was 40% and 53% of the nominal speed, the stability margin decreased by 6.3% and 12.64%, respectively.

The multiphase flow numerical simulation suggests that steam ingestion reduced the incoming Mach number, decreased the kinetic energy, and thickened the boundary layer of most of the rotor suction surface, thus expanding the flow separation area and increasing the wake flow loss. The boundary layer on the suction surface of the rotor was easier to separate under the condition of small mass flow compared with the case of the uniform inlet, which prompted the compressor to enter the stall state ahead of time.

## Figures and Tables

**Figure 1 entropy-22-01416-f001:**
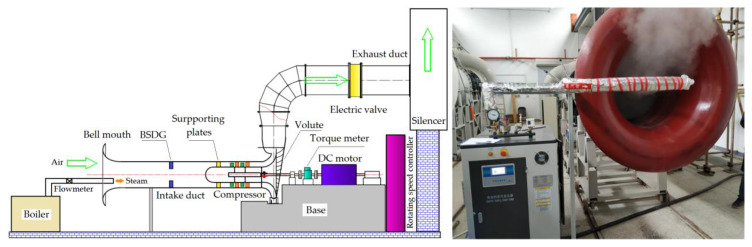
Test bench.

**Figure 2 entropy-22-01416-f002:**
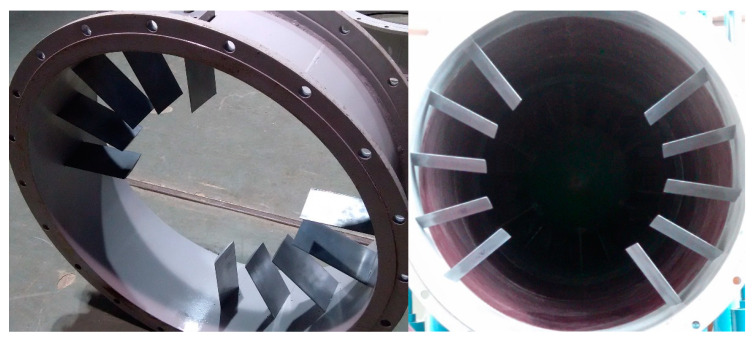
Blade-type swirl distortion generator.

**Figure 3 entropy-22-01416-f003:**
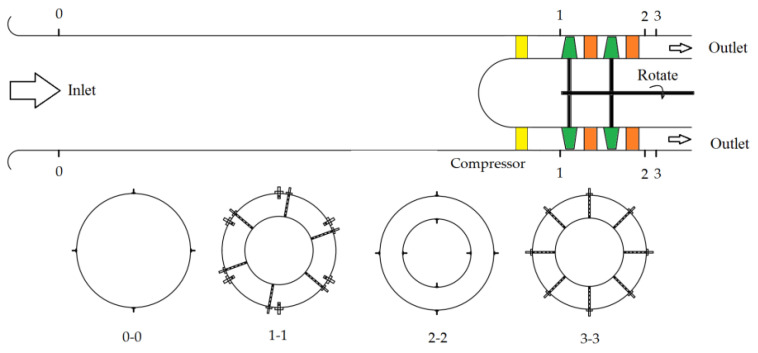
Measuring points layout.

**Figure 4 entropy-22-01416-f004:**
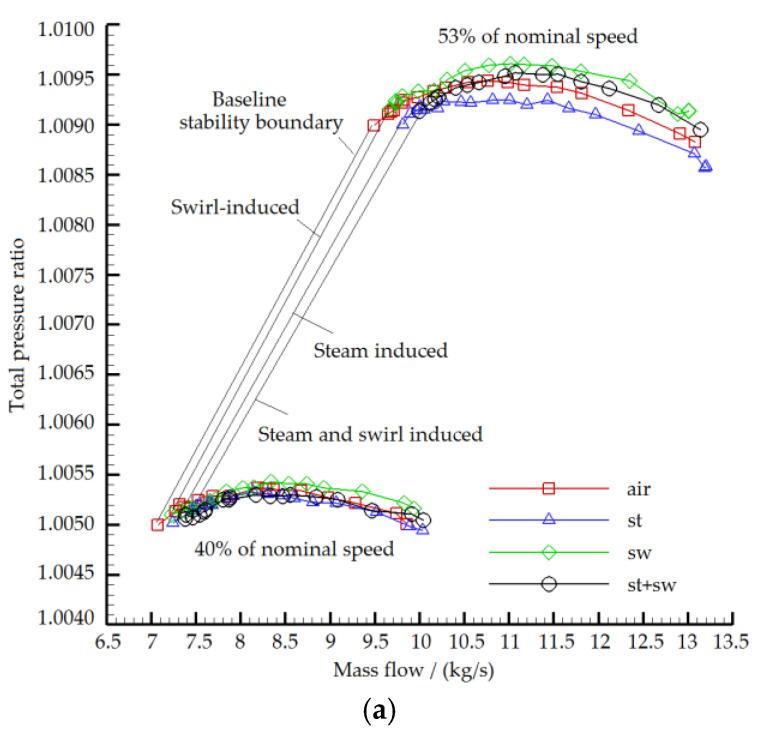
Experimental results of compressor characteristics under different inlet conditions ((**a**) Total pressure ratio vs. corrected mass flow; (**b**) Efficiency vs. corrected mass flow).

**Figure 5 entropy-22-01416-f005:**
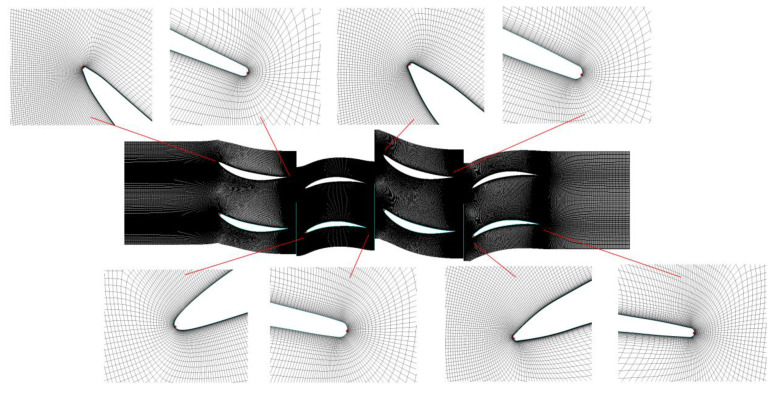
Grid for the single channel.

**Figure 6 entropy-22-01416-f006:**
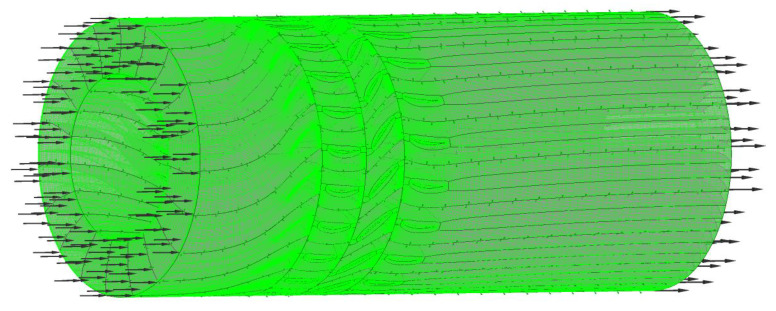
Schematic diagram of the computational grid.

**Figure 7 entropy-22-01416-f007:**
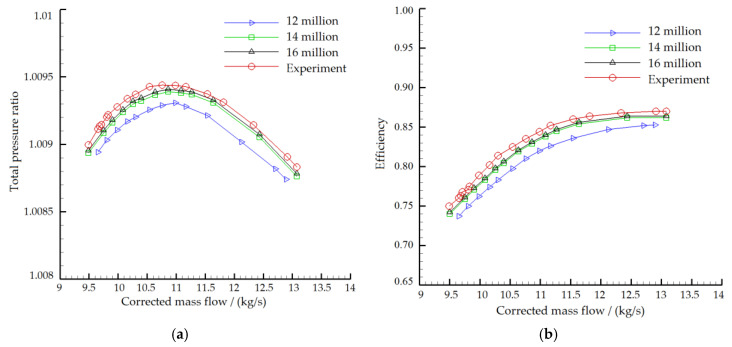
Grid independent verification ((**a**) Total pressure ratio; (**b**) Efficiency).

**Figure 8 entropy-22-01416-f008:**
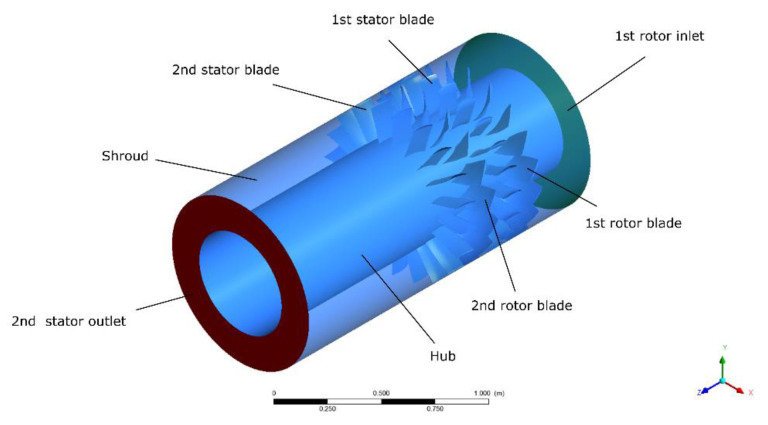
Computational domain.

**Figure 9 entropy-22-01416-f009:**
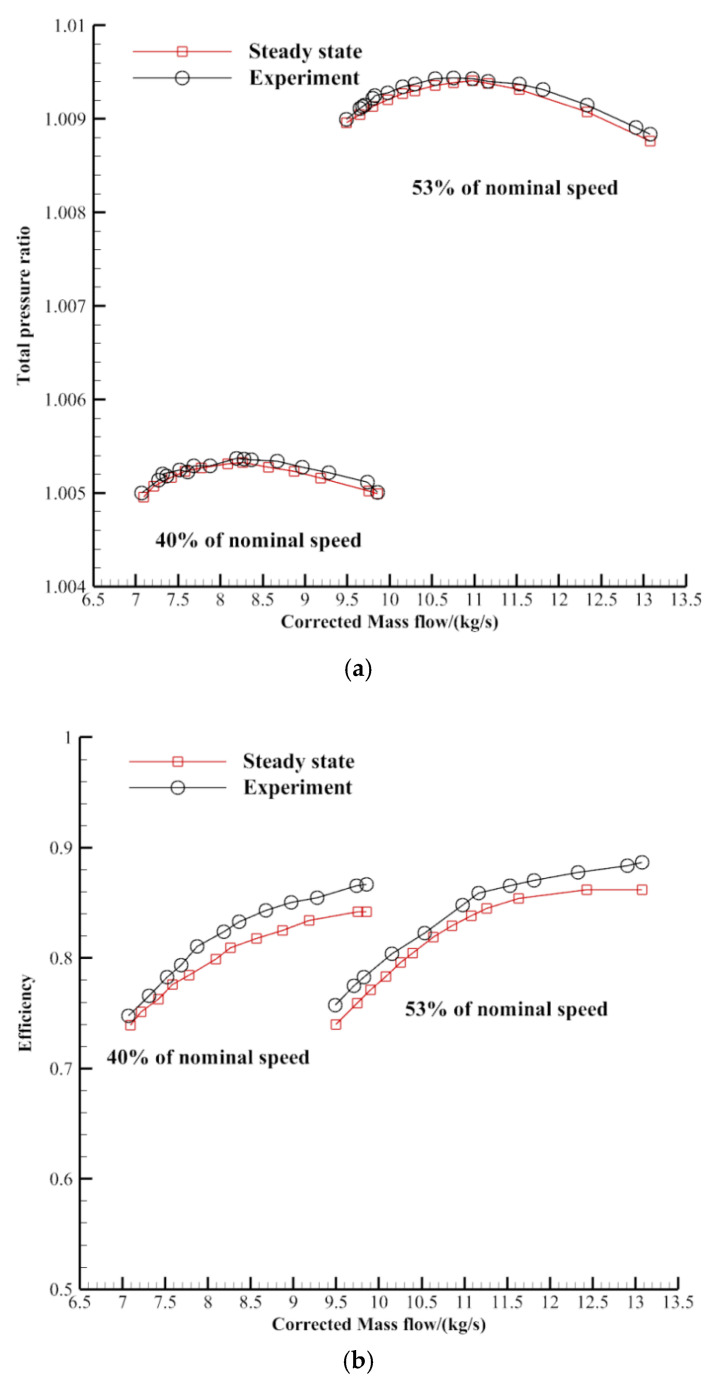
Comparison of steady and experimental characteristic curves ((**a**) Total pressure ratio vs. corrected mass flow; (**b**) Efficiency vs. corrected mass flow).

**Figure 10 entropy-22-01416-f010:**
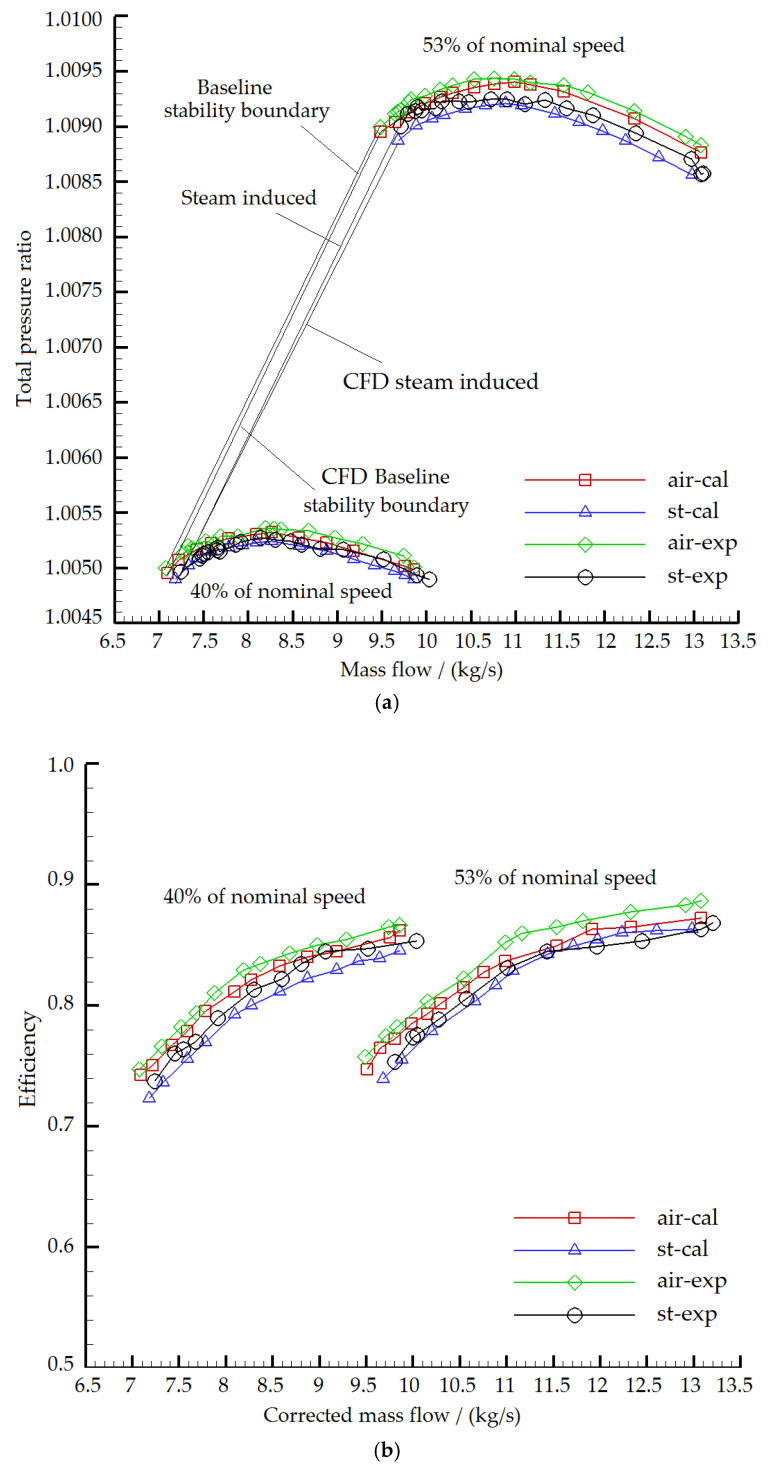
Numerical simulation and experimental results of compressor characteristics under different inlet conditions ((**a**) Total pressure ratio vs. mass flow; (**b**) Efficiency vs. mass flow).

**Figure 11 entropy-22-01416-f011:**
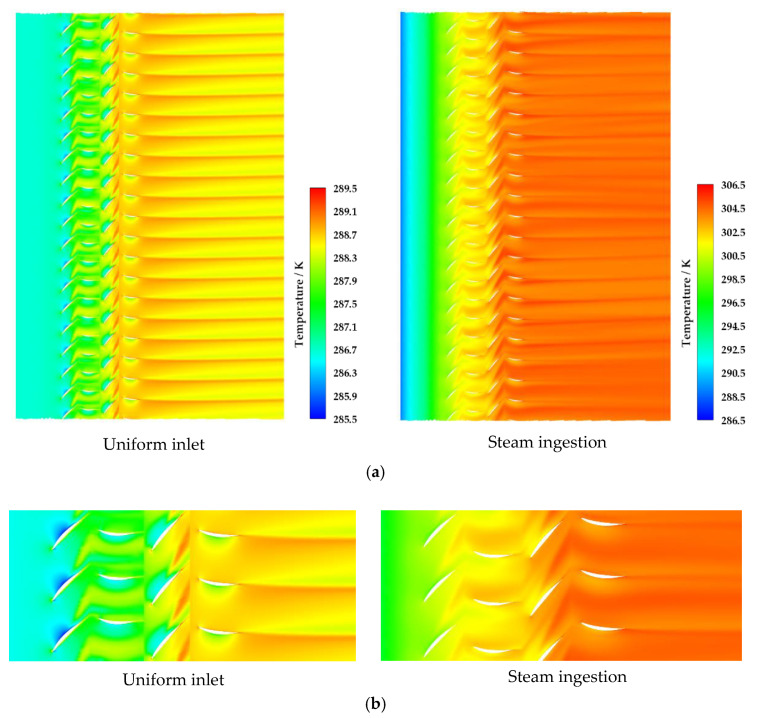
Static temperature distribution of 98% blade height at 53% of nominal speed ((**a**) All passages; (**b**) Local zoom).

**Figure 12 entropy-22-01416-f012:**
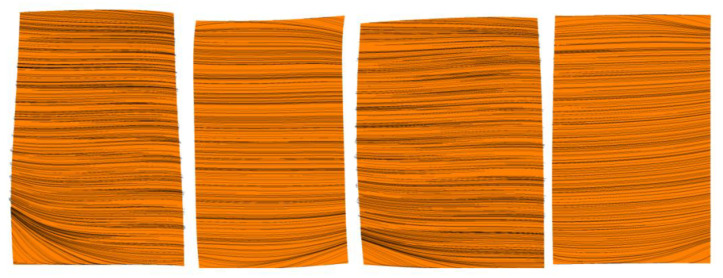
Limiting flow distribution on the suction surface at 53% of nominal speed with uniform inlet flow.

**Figure 13 entropy-22-01416-f013:**
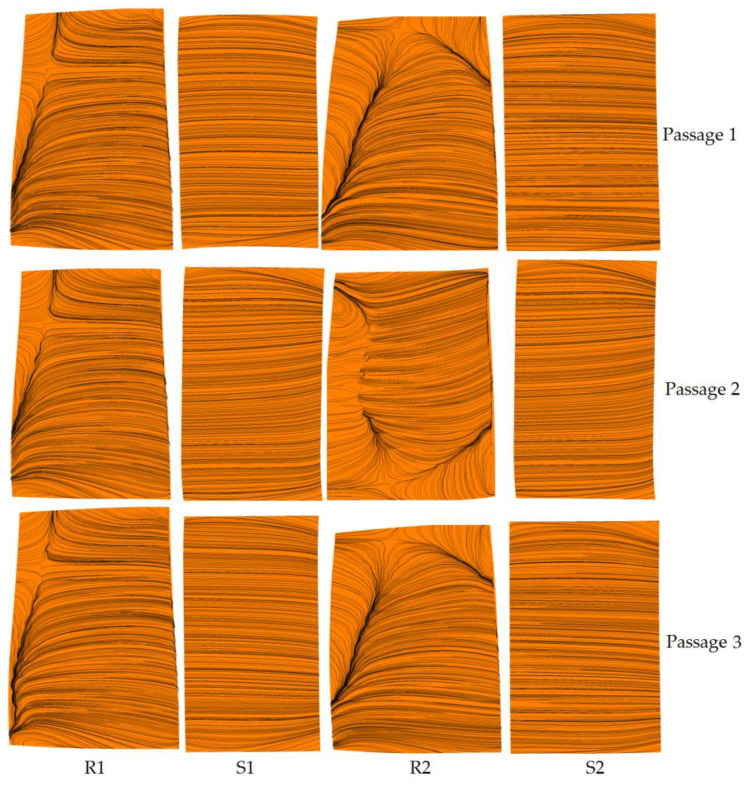
Limiting streamline distribution on the suction surface at 53% of nominal speed with steam ingestion.

**Figure 14 entropy-22-01416-f014:**
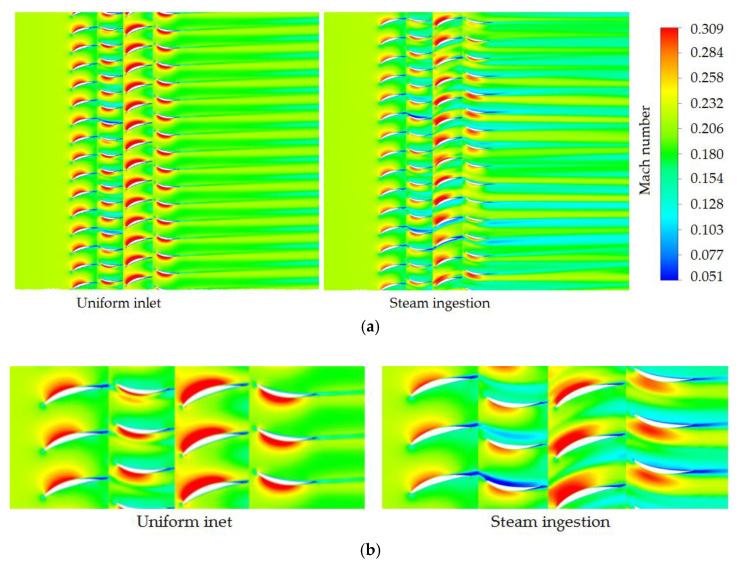
Numerical simulation results of compressor characteristics under different inlet conditions ((**a**) All passages; (**b**) Local zoom).

**Figure 15 entropy-22-01416-f015:**
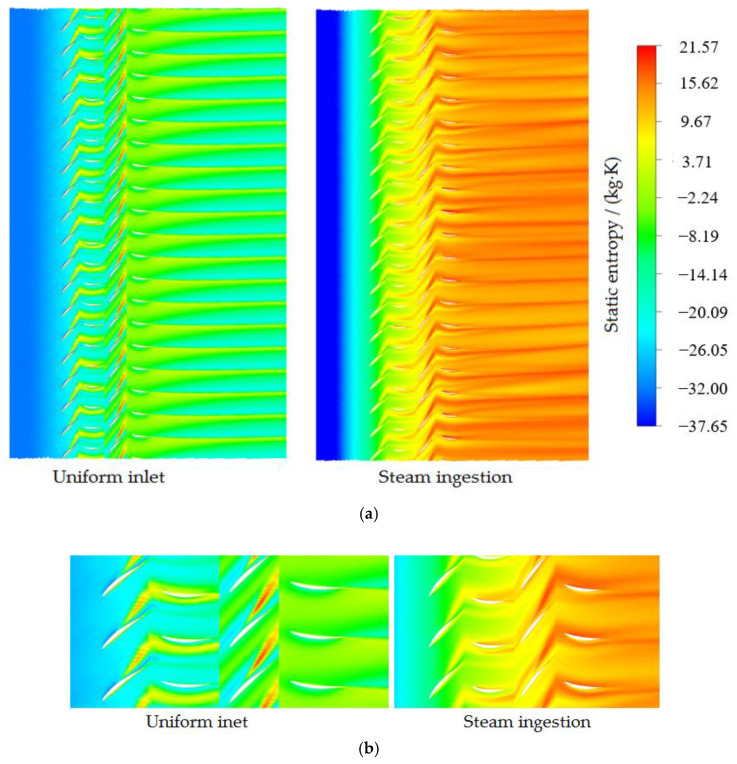
Distribution of air entropy at 98% blade height near the stall point at 53% of nominal speed ((**a**) All passages; (**b**) Local zoom).

**Figure 16 entropy-22-01416-f016:**
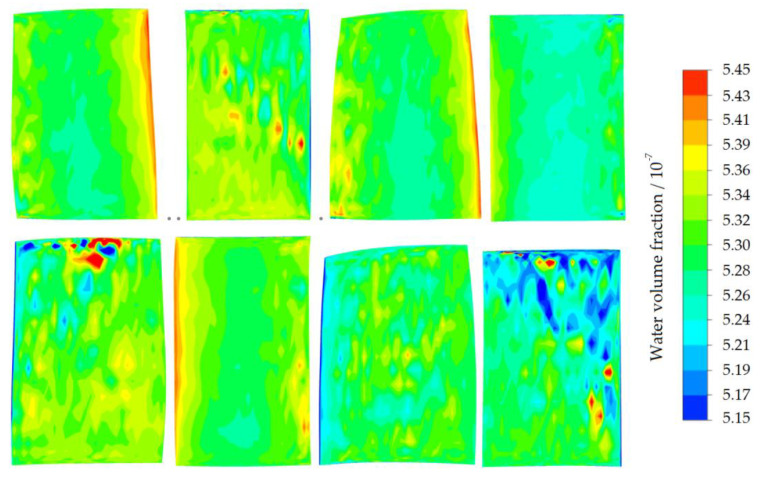
Distribution of water volume fraction on the blade wall under steam suction condition at 53% of nominal speed near the stall point.

**Table 1 entropy-22-01416-t001:** Design parameters of the two-stage low-speed axial compressor.

Parameter	Value			
Outside diameter/mm	900			
Hub-tip ratio	0.6			
Nominal speed/(r/min)	1500			
Total pressure ratio	1.035			
Efficiency	0.88			
Mass flow/(kg/s)	25			
Row	First rotor	First stator	Second rotor	Second stator
Blade profile	NACA-65-010			
Chord length/mm	122	106	130	117
Blade number	19	22	18	20
Radial clearance/mm	1.5	0	1.2	0

**Table 2 entropy-22-01416-t002:** Decrease of the stall margin with steam ingestion.

Rotational Speed	Reference Value/%	Test Value/%	Relative Reduction Value/%
40%	28.25	27.828	1.5
53%	27.45	25.720	6.3

**Table 3 entropy-22-01416-t003:** Decrease in the maximum total pressure ratio with steam ingestion.

Rotational Speed	Reference Value	Test Value	Relative Reduction Value/%
40%	1.005366	1.005301	0.0064
53%	1.00944	1.009242	0.0196

**Table 4 entropy-22-01416-t004:** Decrease of the maximum efficiency with steam ingestion.

Rotational Speed	Reference Value	Test Value	Relative Reduction Value/%
40%	0.866880	0.853439	1.54
53%	0.886811	0.868810	2.03

**Table 5 entropy-22-01416-t005:** Decrease of the stall margin with counter-rotating swirl distortion.

Rotational Speed	Reference Value/%	Test Value/%	Relative Reduction Value/%
40%	28.250	27.240	3.58
53%	27.450	25.688	6.4

**Table 6 entropy-22-01416-t006:** Decrease of the maximum total pressure ratio with counter-rotating swirl distortion.

Rotational Speed	Reference Value/%	Test Value/%	Relative Reduction Value/%
40%	1.005366	1.005398	−0.0032
53%	1.00944	1.009601	−0.0160

**Table 7 entropy-22-01416-t007:** Decrease of the maximum efficiency with counter-rotating swirl distortion.

Rotational Speed	Reference Value/%	Test Value/%	Relative Reduction Value/%
40%	0.866880	0.85049	1.89
53%	0.886811	0.867802	2.14

**Table 8 entropy-22-01416-t008:** Decrease of the stall margin with steam ingestion and swirl distortion.

Rotational Speed	Reference Value/%	Test Value/%	Relative Reduction Value/%
40%	28.250	26.47	6.3
53%	27.450	23.98	12.64

**Table 9 entropy-22-01416-t009:** Decrease of the maximum total pressure ratio with steam ingestion and swirl distortion.

Rotational Speed	Reference Value/%	Test Value/%	Relative Reduction Value/%
40%	1.005366	1.005300	0.0065
53%	1.00944	1.009425	−0.0015

**Table 10 entropy-22-01416-t010:** Decrease of the maximum efficiency with steam ingestion and swirl distortion.

Rotational Speed	Reference Value/%	Test Value/%	Relative Reduction Value/%
40%	0.866880	0.838001	3.32
53%	0.886811	0.850802	4.06

**Table 11 entropy-22-01416-t011:** Boundary condition type.

Zone.	Boundary Condition Type
1st Rotor inlet	Inlet
1st Stator blade	Wall
1st rotor blade	Wall
2nd Stator blade	Wall
2nd Rotor blade	Wall
Hub	Wall
Shroud	Wall
2nd Stator outlet	Outlet

**Table 12 entropy-22-01416-t012:** Comparison of experiment and calculation results of the stall margin reduction value.

Rotational Speed	CFD Stall Margin with Uniform Inlet/%	CFD Stall Margin with Steam Ingestion/%	CFD Relative Reduction Balue/%	Experimental Relative Reduction Balue/%
40%	28.0455	27.145	3.21	1.5
53%	27.6628	25.403	8.12	6.3

**Table 13 entropy-22-01416-t013:** Comparison of experiment and calculation results of the maximum total pressure ratio reduction value.

Rotational Speed	CFD Maximum Total Pressure Ratio with Uniform Inlet	CFD Maximum Total Pressure Ratio with Steam Ingestion	CFD Relative Reduction Value/%	Experimental Relative Reduction Value/%
40%	1.00532	1.00525	0.0080	0.0064
53%	1.00941	1.00919	0.0216	0.0196

**Table 14 entropy-22-01416-t014:** Comparison of experiment and calculation results of the maximum efficiency reduction value.

Rotational Speed	CFD Maximum Efficiency with Uniform Inlet	CFD Maximum Efficiency with Steam Ingestion	CFD Relative Reduction Value/%	Experimental Relative Reduction Value/%
40%	0.86243	0.8454	1.97	1.54
53%	0.88280	0.8624	2.31	2.03

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
