# Peer review of "Experimental and Numerical Investigation on Effects of the Steam Ingestion on the Aerodynamic Stability of an Axial Compressor"

_entropy, 2020, doi:10.3390/e22121416_

Round 1
Reviewer 1 Report
This works is focused I the effects of the steam ingestion on the aerodynamics stability of an axial compressor. The work is carried out by experiments and numerical simulations.
-In my opinion, the paper should be rewritten in a way that the reading would be fluency. The structure of the paper is not good, many information should be reduce (keep the most relevant results) but without losing the most important information. On the other hand, about the quality of the paper which is the novelty of your work?.
—Introduction is not well address.
-Update the references, many of them are old. Also, include prestigious journals (Including entropy) not only proceedings.
-References in line 34 should be changed for a journal.
- Line 42. A Reference of the simulation is needed for the statement.
-Line 61-65, Reference [15] study uniform inlet case and also you, how can you achieve this inlet condition in the experiments?, this condition in the simulation is easy but real conditions should be considered in the simulations.
-Line 107, Is “Tu” a reference, if so reference it correctly.
-Write the equations off the numerical model used.
-Write the equations used to compute the entropy for the air and steam.
-Add a figure of the computational domain and the boundary conditions used.
-The mesh has a lot of cells and the solver was unsteady state, multiphase flow,continuity equation, energy equation and also turbulence thus it is worth to give the information of the CPU or GPU used for the calculations please. Include the total time to solve a case.
-A better figure (high quality) of the mesh should be showed.
-Clearly defy the cases studied and include in the description the operational conditions of the cases.
-In the figure 8, it was expected that the temperature at the inlet were higher in the case of steam ingestion than uniform inlet and then an increase towards the exit. The contours show the inverse. The temperature of the steam is 416.15K and the temperature of the air is 288.15K.
-Write all the figures 8, 11 and 12 in the same time?.
-What about the entropy due to friction and heat transfer?. What kind of entropy is reported in figure 12.
-Line 393-394, It is not clear how the entropy is related with the flow stability.
-About the statements in the lines 412-416 (the same in the conclusion in lines 433-436), i ca not see this behavior in a figure nor in a deep discusión through the paper.
-In the nomenclatura section, add the units please.
Reviewer 2 Report
In this article, the authors perform an experimental and numerical investigation of the two-stage compressor stability during the steam ingestion. The influence of steam ingestion and swirl distortion on the stall margin was studied. The authors should clarify some data. I hope the comments and suggestions presented below may help the authors to increase the quality of the paper:
- Lines 241-243. “Standard k-ε turbulence model based on previous research paper on compressor wet compression and steam ingestion was applied, and the method of scalable wall functions was used near the wall.” Please, provide the reference to the “previous research paper”. It is not clear why the standard k-ε turbulence model was chosen for the study. This model causes unphysical diffusion process in the flow. The clarification is needed why this model was used.
- Figure 8 Why the medium temperature at the inlet for the second case (steam ingestion) is smaller than for the first case? Do the authors use different boundary conditions for temperature?
Lines 380-384 “The separation at the suction surface of the second stage rotor blade was more serious than that of the first stage rotor blade, indicating that the influence of steam ingestion on the second stage rotor was more pronounced. This may be due to the continuous heat transfer between steam and air in the compressor passage, thus entropy and flow loss increased significantly in the passage of the rear stage” What does mean “separation…. was more serious”? How the “continuous heat transfer” causes the separation process? Please clarify. - Figure 12. What is the dimension of air entropy (it is not clear from the annotation)? Which medium state was chosen for 0 value of entropy? Are the 0 (entropy) states the same for Uniform inlet case and steam ingestion case? Why the entropy value at the inlet for the second case in smaller?
Round 2
Reviewer 2 Report
After the revision, the paper became more clear from the scientific point of view, but at the same time new questions arise according to the new text and authors` responses:
1. A small comment according to the authors' response to my previous remark about the standard k-epsilon model. I am not sure about the fact that this model “is viewed as a normal industrial turbulence model in the industry community”. Its modifications (for example, k-epsilon realizable model) are more accurate and widely used. The authors provided the reference to the previous work, where the CFD simulation using this model was conducted. According to presented in this reference work results, the CFD prediction (in comparison with experiment) was not so good in my opinion. I agree that such a model may be used, but the motivation is not clear for me. I think if the authors used a more advanced model (this does not necessarily lead to a significant slowdown in the calculation) the results could be more accurate and clear.
2. Again the figure 15. Please take a look at the area near the color map. The dimension of entropy is wrong. It should bi J/(kg*K).
3. Lines 276-278 “The discrete phase governing equations of liquid droplets in steam were solved by Lagrangian method. The discrete governing equations include the motion equation of droplets of liquid water heat transfer equation, and mass transfer equation. The specific equations are shown below. Where is the mass transfer equation? The equation 12 corresponds to saturation pressure estimation.
4. The authors used DPM to calculate the water droplets movement. Is there any information about droplets diameters spectrum at the inlet boundary? Are their sizes correspond to the experimental conditions?
5. I am lost track with figure 11. The authors' response to my question according to figure 11: “The medium temperature under the first case (steam ingestion) was higher than that under the second case (mere air ingestion). The temperature distribution in the second case was broader than the first case, and they were drawn in on the same scale for comparison. Therefore, the color gradients seemed distinct in the second case” . But the first case (a) is “uniform inlet” and the second case is “steam ingestion” (according to the text under the figure) Is it a mistake in the paper? So, if the “a” figure is steam ingestion and “b” figure is the condition without the steam, then why the medium temperature of only air (without steam) grows upstream the first stage? Is there a heat transfer phenomena? Please clarify in more detail! For me, it seems a nonphysical result, but I may be wrong due to a very complicated explanation.
6. A subjective comment which may not be counted during the paper revision. The authors added in the text the equations of used models (equations 7 – 12). A short comment about used models was provided. These equations look in this paper excess. Because the authors didn`t add used letters in nomenclature, there is no any comment of the way the mixture thermodynamics parameters were calculated, which forces acting the water droplets were taken into account and so on. It seems the authors provide these equations “just to be”.
Author Response
Please see the attachment.

This manuscript is a resubmission of an earlier submission. The following is a list of the peer review reports and author responses from that submission.
Round 1
Reviewer 1 Report
This paper aims at investigating the effect of steam ingestion on the stability of a low speed two-stage axial compressor. Experiments and numerical simulations are compared for several inflow conditions. Expected behaviors are found both experimentally and numerically, namely a decrease of stability margin for steam ingestion configurations. The originality of this research work lies in the use of a multiphase flow simulation to model the mixture of air and steam, whereas existing researches used a gas mixture as a unique fluid. This point thus justifies the publication of this paper as a research journal paper. However, some major remarks detailed below must be addressed before considering a possible publication of this paper.
Major remarks
- The compressor is investigated experimentally and numerically at 600 and 800 RPM, whereas the nominal rotational speed is 1500 RPM. These numbers are very specific to this compressor, and do not make sense for the reader. I strongly suggest replacing these numbers by the percentage of the nominal rotational speed, i.e. 40% and 53% respectively in the whole paper. The compressor is thus run at part-speed, meaning that its aerodynamic behavior is very different from running at nominal speed. The conclusions proposed in the paper can not thus be easily applied to a compressor running at nominal speed. For instance, the stall mechanism at 40% and 53% of nominal speed is probably not the same at full speed. This point should be largely commented in the paper.
- The description of the numerical simulations should be detailed more precisely. Regarding the computational domain, what the inlet and outlet boundary conditions? Are they representative of the experiments? Are all the simulations performed with a 360 degrees configuration? How many blades per row are present in the simulation? Are the rotor tip gaps taken into account in the simulation? What are the numerical rotor/stator interfaces, for steady and unsteady calculations? Regarding the grid: how many points per blade passage, and per rotor tip gaps if they are simulated? What are the resulting y+ values on blade surfaces? Is the grid properly designed for simulations?
- The distinction between steady and unsteady simulations is not clear for the reader. A comparison is performed in Fig. 5, showing little effect of unsteady parameter. It is not proven in the paper that the stability boundary point is predicted more accurately with unsteady simulations than with steady simulations. Only a comparison to experimental data could prove this point. This crucial point must be corrected.
- Are all the simulations shown in section 6 unsteady? This is not clear for the reader what are the steady and the unsteady simulations. Maybe a table would help the reader to better understand this.
- How is defined the stability limit in simulations? Since the last stable point obtained is at the core of the discussions in the paper, some care must be taken. Indeed, the stall margin estimated in the simulations is highly dependent of the definition of the last stable point. The uncertainty of the numerical stall margin is expected to be significant. Stall margins can thus be compared relatively between different simulations, but their absolute comparisons with experimental data is not straightforward. This point should be discussed with care in the paper.
- Equation 5 refers to the definition of corrected mass flow. Is this corrected mass flow used for Figures 6 and 7? It should since the inlet conditions are not the same across the configurations that are compared. If not, Figures 6 and 7 must absolutely be corrected.
- Figure 7 compares experimental and numerical results. Because of the scales, it is difficult to clearly see these comparisons for the reader. It would be relevant to add other figures relative to a given speed, thus bringing a zoom on these comparisons.
- Figures 11 and 12 present a lot of blades, preventing the reader to clearly see the details of the flow, for instance in boundary layers and wakes. It would be interesting to add other figures, zooming on one or two successive blade passages.
- Could the authors precise how is computed the entropy in a multiphase context, and how does it differ from entropy for a single phase calculation?
- Figure 13: the colors lie in the green-yellow range on the figure, whereas the colorscale goes from blue to red. Could the authors refine the scale, to improve the readability of the figure?
- How do the authors explain that pressure ratio increases with a swirling inflow condition? Is it a consequence of the machine running at part-speed, would it be the same at full speed?
- Even if the originality of the paper is to present multiphase CFD simulations for configurations with steam ingestion, it is not clear what breakthrough this modelling brings with respect to existing literature on the subject. I suggest the authors complete their analysis in showing why it would be relevant for others to use a similar numerical approach to investigate such configurations.
Minor remarks
- Line 36: Pennsylvania should probably be replaced with California
- Line 114: Figure 3 should be replaced with Figure 2
- Figure 7: the title is misleading, it should mention numerical and experimental results
- Table 5: typo mistake in the third column (0. should be removed)
Author Response
We have revised our manuscript entitled " Experimental and Numerical Investigation on Effects of the Steam Ingestion on the Aerodynamic Stability of an Axial Compressor " submitted to “entropy” for publication. The ID is entropy-931304. Many thanks to all the experts for their comments on this paper. According to the comments, the author has made a corresponding revision to the paper. Please see the attachment.

Reviewer 2 Report
In this article, the authors provide an experimental and numerical investigation of the two-stage compressor stability during the steam ingestion. The influence of steam ingestion and swirl distortion on the stall margin was studied.
Unfortunately, the way the authors presented their results doesn`t allow to effectively judge the correctness of obtained data. The paper should be reworked. I hope the comments and suggestions presented below may help the authors to increase the quality of the paper:
- Section 2 Experimental equipment. This section doesn`t provide complete information about the experimental rig. The scheme with the main sizes is needed, the geometry of the compressor flow path is needed.
- Section 3 Experimental method. The measurement methodology is unclear. Which parameters and where were measured? Which measurement devices were used? What are their measurement errors?
- During the reading the manuscript, I was confused about the steam conditions. Is it superheated or saturated or wet? According to the figure 1, it is wet; according to the sentence on lines 145-146 “In order to highlight the change of inlet total temperature caused by steam ingestion, the existence of a few liquid tiny diameter water droplets in the steam was ignored“ it is superheated; according to the sentence on lines 214-215 “Considering that the steam generated by the steam generator contained water droplets, the mass fraction of liquid water was set as 1/10 of the gaseous steam…” the steam is wet with the wetness 10%. I think the clarification is needed to resolve this misunderstanding.
- Please, provide the clarification of the reason the steam temperature and droplets temperature are so different (lines 217-219)? Is it a feature of the steam generation? The two-phase medium is not in the thermodynamic equilibrium state. As a result, the active steam condensation should proceed. Was it taken into account during the simulation? Which condensation model was used? How these temperatures were measured at the experiment?
- Lines 181-189. Unfortunately, I am not sure that I clearly understood the text. Maybe some text reworking may increase the readability.
- The sentence on lines 215-217 “The k-ε turbulence model based on previous research paper on compressor wet compression and steam ingestion was used.“ please provide the reference to the paper. What type of k-ε model was used? Which wall function was used?
- The information presented in figure 5. Which rotor-stator interaction model was used? How the rotation was taken into account for steady calculation?
- Section 5. For me not clear why the authors in detail describe the parameters change in percent. For another experimental rig, these values will be another. From the other side, the authors don`t provide any physical interpretation of the obtained data. The mechanism of the steam ingestion influence is not explained.
- Section 6. Please clarify the significant of parameters variation about 0.0016%, 0.0080%, etc. (lines 312-315). Is it an important change? Maybe this is just a CFD deviation. Maybe it is very important for the regime point near the stall. But the clarification is needed.
- Figure 8 generates several questions
- Why do the authors compare the steady results for uniform inlet and instantaneous temperature distribution of unsteady calculation?
- The sentence on lines 329-321 ” The air static temperature of the flow increased continuously from the inlet of the compressor due to steam ingestion, and the temperature rise was especially noticeable after the flow pass through the second stage rotor passages” What does mean “temperature rise was especially noticeable”? According to the colormap (from 294.4 to 305.5), the change doesn`t exceed 3 degrees. Please clarify why it is a noticeable change?
- Why the temperature at the inlet for the steam case is lower than for the uniform inlet? According to the authors, the steam temperature was 416 K, so the mixture temperature for the steam case at inlet should be bigger.
- The sentence on lines 349-350 “This may be due to the continuous heat transfer between steam and air in the compressor passage”. Please provide the physical interpretation.
- Figures 9 and 10. Which operating point (on figure 7) provided figures correspond?
- The conclusions. Again – the change of parameters by 0.0032% and 0.016% (see 9).
Author Response

(The authors gave the same response as above.)

Reviewer 3 Report
This works is focused I the effects of the steam ingestion on the aerodynamics stability of an axial compressor. The work is carried out by experiments and numerical simulations.
-In my opinion, the paper should be rewritten in a way that the reading would be fluency. The structure of the paper is not good, many information should be reduce (keep the most relevant results) but without losing the most important information. On the other hand, about the quality of the paper which is the novelty of your work?.
—Introduction is not well address.
-Update the references, many of them are old. Also, include prestigious journals (Including entropy) not only proceedings.
-References in line 34 should be changed for a journal.
- Line 42. A Reference of the simulation is needed for the statement.
-Line 61-65, Reference [15] study uniform inlet case and also you, how can you achieve this inlet condition in the experiments?, this condition in the simulation is easy but real conditions should be considered in the simulations.
-Line 107, Is “Tu” a reference, if so reference it correctly.
-Write the equations off the numerical model used.
-Write the equations used to compute the entropy for the air and steam.
-Add a figure of the computational domain and the boundary conditions used.
-The mesh has a lot of cells and the solver was unsteady state, multiphase flow,continuity equation, energy equation and also turbulence thus it is worth to give the information of the CPU or GPU used for the calculations please. Include the total time to solve a case.
-A better figure (high quality) of the mesh should be showed.
-Clearly defy the cases studied and include in the description the operational conditions of the cases.
-In the figure 8, it was expected that the temperature at the inlet were higher in the case of steam ingestion than uniform inlet and then an increase towards the exit. The contours show the inverse. The temperature of the steam is 416.15K and the temperature of the air is 288.15K.
-Write all the figures 8, 11 and 12 in the same time?.
-What about the entropy due to friction and heat transfer?. What kind of entropy is reported in figure 12.
-Line 393-394, It is not clear how the entropy is related with the flow stability.
-About the statements in the lines 412-416 (the same in the conclusion in lines 433-436), i ca not see this behavior in a figure nor in a deep discusión through the paper.
-In the nomenclatura section, add the units please.
Round 2
Reviewer 1 Report
The explanation of the computational method has been improved in the second version of the paper, however it is not precise enough for publication in a journal paper.
Y+ values are correct, please add them in the paper.
Rotor tip gaps are not simulated. This is a significant simplification for a turbomachinery CFD simulation and it should be absolutely mentioned in the paper. Moreover, it is known that the flow in the rotor tip regions has often a large influence on stall mechanisms. Could the authors prove that this compressor is insensitive to the tip flows close to stall?
Validation of simulations are performed using only global integrated values. A step further in the validation would be to compare radial profiles with experimental values in order to gain confidence in the computations.
The authors explain that steady computations are performed with the frozen rotor approach, and the unsteady computations with the mixing plane approach. This is not consistent with Fig. 8 and 11, where the figures show a mixing plane computation for a uniform inlet, and a frozen rotor computation for steam ingestion. There may be an error in the text.
Fig. 8, 11 and 12 should contain comparisons of similar approaches, i.e. only mixing plane simulations. Present comparisons are not relevant.
From what I understand, the only unsteady simulated phenomenon is the continuously varying back pressure in the mixing plane simulation. The term unsteady is misleading since rotor-stator interactions are not taken into account. Could the authors explain that more clearly in the paper?
Reviewer 2 Report
The paper looks good. Currently, I have 2 comments according to the corrected article:
- The author's response (Response 10a) according to figure 8: “… was corrected as “including the unsteady calculation results with uniform inlet and the unsteady calculation results with steam ingestion (t = 1.998s)”. But the text on lines 352-353 was not corrected in the submitted paper: “including the steady calculation results with uniform inlet and the unsteady calculation results with steam ingestion (t = 1.998s)”
- I am not completely satisfied with the author's response according to the variation of the values in a range of 0.0016%, 0.008%. It is a very small deviation – experimentally it is impossible to highlight such difference. In CFD the truncation error brought in results also (besides the mesh and solver options) depends on absolute values and their gradients in a computational mesh. The authors in their response don’t explain the significance of such deviation. My point – such a difference in flow parameters shouldn’t be considered in the paper especially in conclusions. I might be wrong, but the author's clarification is needed.